# Endothelial Cell Dysfunction Due to Molecules Secreted by Macrophages in Sepsis

**DOI:** 10.3390/biom14080980

**Published:** 2024-08-09

**Authors:** Heng He, Wei Zhang, Luofeng Jiang, Xirui Tong, Yongjun Zheng, Zhaofan Xia

**Affiliations:** 1Department of Burn Surgery, The First Affiliated Hospital of Naval Medical University, Shanghai 200433, China; heheng2028@163.com (H.H.); 15821680963@163.com (W.Z.); jiangluofeng1999@163.com (L.J.); ruirui_gogo@163.com (X.T.); 2Research Unit of Key Techniques for Treatment of Burns and Combined Burns and Trauma Injury, Chinese Academy of Medical Sciences, Shanghai 200433, China

**Keywords:** sepsis, macrophage, endothelial dysfunction, inflammatory factor, adhesion, permeability, treatment

## Abstract

Sepsis is recognized as a syndrome of systemic inflammatory reaction induced by dysregulation of the body’s immunity against infection. The multiple organ dysfunction associated with sepsis is a serious threat to the patient’s life. Endothelial cell dysfunction has been extensively studied in sepsis. However, the role of macrophages in sepsis is not well understood and the intrinsic link between the two cells has not been elucidated. Macrophages are first-line cells of the immune response, whereas endothelial cells are a class of cells that are highly altered in function and morphology. In sepsis, various cytokines secreted by macrophages and endothelial cell dysfunction are inextricably linked. Therefore, investigating how macrophages affect endothelial cells could offer a theoretical foundation for the treatment of sepsis. This review links molecules (TNF-α, CCL2, ROS, VEGF, MMP-9, and NO) secreted by macrophages under inflammatory conditions to endothelial cell dysfunction (adhesion, permeability, and coagulability), refining the pathophysiologic mechanisms of sepsis. At the same time, multiple approaches (a variety of miRNA and medicines) regulating macrophage polarization are also summarized, providing new insights into reversing endothelial cell dysfunction and improving the outcome of sepsis treatment.

## 1. Introduction

The latest definition of sepsis is a critical condition of disordered immune reaction triggered by infection [1]. Common sources of infection are the lungs, abdomen, and urinary system [2]. Sepsis-related deaths account for about 20% of all deaths worldwide [3]. One of the characteristics of sepsis is the disruption of the endothelial barrier due to systemic inflammatory response syndrome. Endothelial cell dysfunction is unequivocally the mechanism of sepsis [4]. However, there are still many research gaps in how macrophages affect endothelial cells in sepsis. In the physiological condition, macrophages are present in the bloodstream mainly in the form of monocytes [5,6]; endothelial cells exchange substances with tissues at the level of microvessels and maintain functional homeostasis of organs. In sepsis, macrophages perform the duties of removing pathogens and triggering inflammation [7]; in turn, endothelial cells act as a fence between tissues and the bloodstream, tending to undergo morphological and functional changes in the modulation of permeability, coagulation, and inflammatory response due to endotoxins and cellular factors [4]. Macrophage–endothelial cell interactions have been well-documented in a number of important diseases such as atherosclerosis [8]. Similarly, the interplay of these two cell types in sepsis as well as the restoration of the endothelial barrier and the modulation of macrophage activation for the treatment of sepsis are of increasing interest to scholars.

In order to reduce sepsis mortality, to explore the pathophysiologic mechanisms of sepsis and to find more potential therapeutic approaches, it makes more sense to discuss the specific mechanisms of the interaction of multiple cells. Therefore, this paper reviews the mechanisms of endothelial cell and macrophage interactions with the aim of providing new insights and perspectives on the pathophysiologic mechanisms and treatment of sepsis.

## 2. Basic Functions of Macrophages and Endothelial Cells

Macrophages are a vital part of human innate immunity, which were originally found by Elie Metchnikoff, for which he won the Nobel Prize in 1908 [9]. The precursor cells of macrophages are monocytes, which mainly originate in the marrow, are transported through the body’s circulation, and eventually mobilize to the tissues and differentiate into macrophages [5,6]. The main functions of macrophages include phagocytosis, antigen presentation, and secretion of cytokines including multiple interleukins (ILs) and tumor necrosis factor (TNF)-α [7]. Nevertheless, a growing body of research demonstrates that macrophages not only perform a function in immune defense but are more often associated with other cells and thus contribute to the regulation of homeostasis in the body. The idea that macrophages are capable of polarization was proposed in 1992 [10]. Studies on macrophages have shown that they are mainly categorized into the M1 type (pro-inflammatory) and M2 type (anti-inflammatory) [11]. The polarized macrophage can perform regulatory functions in a wider range of tissue microenvironments, which highlights the superior plasticity that macrophages possess.

In a normal organism, the vascular endothelial barrier consists mainly of the extracellular matrix, several kinds of intercellular junctions, and glycocalyx [12]. This barrier is important for maintaining plasma colloid osmotic pressure because it retains macromolecules, such as proteins, in the blood and is a prerequisite for preventing tissue edema. In some organs, endothelial cells can form special selective barrier structures with neighboring cells to maintain organ homeostasis. In the lungs, endothelium and other lung tissue cells form the well-known air–blood barrier for efficient gas exchange [13]. Such structures are also observed in organs such as the brain. In addition to this, healthy endothelial cells also act as smooth anticoagulant surfaces to allow for a steady flow of blood [14].

## 3. Macrophages Secrete a Variety of Substances to Wake Up Endothelial Cells

In the setting of sepsis (infections triggered by burns, trauma, etc.), endothelial cells are activated by the stimulation of pathogens and other cellular products secreted by macrophages, which will alter their normal structure and function [15]. The receptor causing these alterations is primarily Toll-like receptor-4 (TLR4) [16]. Among these changes, increased adhesion promotes blood cell aggregation, which in turn exacerbates the inflammatory response and endothelial cell damage. Meanwhile, exudation of intravascular fluid caused by enhanced vascular permeability is also a significant hallmark of septic shock. Fluid leakage is not only a manifestation of inflammation, which causes insufficient circulating blood volume and organ hypoperfusion, but more importantly, subsequent fluid resuscitation may exacerbate tissue edema [17]. Coagulation disorders are also an important feature of sepsis. Therefore, we will next discuss how macrophages alter endothelial cell adhesion, permeability, and coagulability under septic conditions (as shown in Figure 1). We also summarize the functions and possible mechanisms of the main molecules discussed, as shown in Table 1.

### 3.1. Macrophages Increase Endothelial Cell Adhesion and Ability to Recruit Leukocytes

Under normal physiological conditions, a small proportion of macrophages reside in some organs and have proprietary names, including alveolar macrophages (lungs, also called dust cells), microglia (brain), and kupffer cells (liver) [18]. So when pathogens invade, it is still necessary for monocytes (precursor cells of macrophages) in the bloodstream to move directionally to the blood vessels at the site of infection. When tissue-resident immune cells, including macrophages, are activated, they also secrete pro-inflammatory cytokines, which can activate endothelial cells into a pro-inflammatory phenotype to promote leukocyte efflux (monocytes, macrophages, and neutrophils) and exudation of plasma components [19]. The phenotypic transformation of endothelial cells in the presence of inflammatory agents and bacterial products is inextricably associated with the nuclear transcription factor kappa-B pathway (NF-κB) [20]. Lipopolysaccharide (LPS) can further activate NF-κB after binding TLR4 on the endothelial cell surface [21]. Activated endothelial cells will secrete some chemokines to increase their recruitment for leukocytes, including monocytes [22]. Among them, monocyte chemotactic protein-1 (also known as CCL2) belongs to a representative class of chemokines. It is the first molecule in the human CC chemokine subgroup to be purified and isolated and exhibits a favorable monocyte chemotactic effect in vitro [23]. Its main sources in the body are microglia, endothelial cells, and monocytes/macrophages [24]. The liver is the parenchymal organ that contains the largest number of macrophages in the organism and is also one of the organs that contains the most amount of blood [25]. Some studies have shown that every tenth hepatocyte is accompanied by 2–4 macrophages [26]. However, even with such a high proportion of endogenous macrophages, when inflammation occurs in the liver, large numbers of marrow-originated mononuclear cells are recruited to the site of injury in a targeted manner, relying on CCL2 and its receptor [27]. This suggests the importance of CCL2 in the recruitment of monocytes by endothelial cells, and it is worth noting that both monocytes and endothelial cells can secrete CCL2 and therefore may lead to a positive feedback recruitment effect of monocytes/macrophages, triggering excessive inflammatory cell infiltration and causing tissue damage.

Stimulated by inflammatory factors, endothelial cells also highly express some adhesion molecules on their surface in addition to secreting extracellular chemotactic substances. One group of substances, called selectins, is thought to play a mediating role in the adhesion of endothelial cells to monocytes. Selectins are broadly categorized into three groups, including E-, L-, and P-selectin [28]. Selectins maintain low expression on endothelial cells but are only strongly upregulated by inflammatory factor stimulation through the NF-κB pathway [29]. However, because humans lack transcriptional regulatory elements for P- and L-selectin, they predominantly increase E-selectin expression in response to stimulation by LPS and inflammatory cytokines produced by macrophages [30]. Similarly, the LPS-mediated generation of inflammatory factors also upregulates several adhesion molecules including intercellular adhesion molecule-1 (ICAM-1) and vascular cell adhesion molecule-1 (VCAM-1) via the NF-κB pathway on the endothelial cell surface [31]. It has been shown that TNF-α is able to increase the expression of two adhesion molecules through the ZIPK (zipper-interacting protein kinase)/NF-κB signaling axis [32]. In addition to this, endothelial cells can release extracellular vesicles containing VCAM-1 that bind to receptors on monocytes, shifting them toward a pro-inflammatory state [33]. This may exacerbate inflammatory responses. When converging incoming leukocytes reach the vicinity of the inflamed area, the endothelial cell’s E-selectin can recognize glycan counter-receptors expressed by monocytes, causing it to roll slowly over the surface of blood vessels in the high velocity of blood flow [34]. Meanwhile, integrins on the surface of monocytes can also bind to two kinds of adhesion molecules exposed on the luminal side of endothelial cells, further stabilizing the adhesion between the two types of cells [34,35]. Finally, monocytes traverse the activated endothelial cell layer via the paracellular pathway (transient opening of intercellular junctions) as well as the transcellular pathway (passage within the endothelial cell) [36,37]. However, recent research has discovered that E-selectin may also be engaged in negatively regulating endothelial cell adhesion. In the in vitro setting of LPS, knockdown of E-selectin downregulated actin activation and destabilized endothelial cell junctions, ultimately enhancing leukocyte adhesion and migration to endothelial cells [38]. However, this study mainly used an in vitro model; thus, further exploration of the overall effects in vivo is needed.

### 3.2. Macrophages Increase Endothelial Cell Permeability in Various Ways

Multiple organ dysfunction syndrome (MODS) belongs to the common complications of sepsis. MODS occurs in approximately 50% of the patients in the ICU and is associated with death in 30% of patients [3,39]. The development of MODS is closely associated with the permeability of the endothelial barrier in the microcirculation. Take the lung, for example; it is one of the susceptible target organs in sepsis. One of the important reasons for this is its very rich microvasculature for gas exchange with the external air. In addition to this, the lungs, as organs that interact directly with the external environment, contain a large number of cells of innate immunity including macrophages. Acute respiratory distress syndrome (ARDS) is considered the most critical form of lung injury, accounting for approximately 10% of ICU patients [40]. The pathological feature of ARDS is diffuse alveolar injury: alveolar edema due to increased capillary permeability, massive inflammatory cell infiltration, and necrotic detachment of the alveolar epithelium, ultimately leading to dysfunctional gas exchange [41]. Therefore, it is likely that macrophages with pro-inflammatory effects modulate the permeability of vascular endothelial cells in MODS through multiple pathways, which influences the prognosis of septic patients. Normal permeability structures are organized by glycocalyx, cell-to-cell junctions, and their skeleton and intact endothelial cells. Hence, in the next chapters, we are going to analyze several possible ways in which molecules secreted by macrophages may affect these structures to increase endothelial cell permeability.

#### 3.2.1. Injured Gatekeepers—Glycocalyx of Endothelial Cells

The concept of glycocalyx dates back to 1963, when it was used to describe the polysaccharide coating on the outside of endothelium [42]. More studies later revealed the more subtle composition of the glycocalyx. The glycocalyx consists of three main components, roughly including proteoglycans, glycosaminoglycans, and plasma proteins [43]. In acute injury and possible subsequent infectious sepsis, the glycocalyx becomes the earliest structure of the endothelium to be affected, which is why it is also called the gatekeeper. When enzymatic removal of the glycocalyx is used, the isolating effect of the vascular barrier is reduced, leading to a redistribution of fluid on both sides of the vessel [44]. Apart from this, the surface of the glycocalyx has a high density of negative charges that repel the predominantly negatively charged proteins in the blood, thus not allowing albumin to pass through [45]. Therefore, the endothelial cells and the glycocalyx work together to maintain an integral vascular barrier. The normal glycocalyx of the organism has the ability to rapidly restore its homeostasis. It has been found that the glycocalyx, which is acutely shed due to exposure to shear stress and enzymes, is able to regain its original thickness in 5 to 7 days [46]. Another ex vivo research study revealed that glycocalyx recovery on the surface of endothelial cells could even be accomplished within 20 h [47]. A theory has been proposed for the mechanism of quick restoration of the glycocalyx in a non-septic environment, which is dependent on fibroblast growth factor receptor 1 (FGFR1) and heparin sulfate biosynthetic enzyme (EXT1). However, in a septic environment, the FGFR1/EXT1 signaling axis is partially inhibited [48]. Inflammatory proteins produced by macrophages might be associated with delayed restoration of the endothelial glycocalyx. By intraperitoneally injecting FGF2, a member of the FGF family, into septic mice with cecum ligation and puncture (CLP), the researchers found a reduction in capillary leakage and macrophage infiltration in the mice’s lung tissues, as well as an increase in survival [49]. This could be a potential target for restoring glycocalyx structure.

After endothelial cells are activated by pro-inflammatory factors released by macrophages in sepsis, the glycocalyx disintegrates and falls off, decreasing in thickness. In a mouse model of interstitial pulmonary edema, the researchers used an electron microscope to observe the breakdown of the glycocalyx. Apart from this, an increase in syndecan-1 (SDC-1), a marker of glycocalyx damage, was also detected. They further removed macrophages from the model and found that the pulmonary edema was alleviated and glycocalyx degradation was inhibited [50]. This suggests that macrophages might be associated with the regulation of permeability by the glycocalyx. TNF-α may perform a key function in this process. Some studies have shown that in sepsis-related lung injury, TNF-α produced by macrophages is able to activate heparanase, which degrades heparin sulfate (a crucial part of the glycocalyx) [51]. Apart from this, research has also revealed that TNF-α enhances the expression of matrix metalloproteinase 9 (MMP-9), which can lead to the detachment of glycocalyx structures from the glomerular endothelium [52]. In an acute lung injury (ALI) model in mice caused by LPS tracheal nebulized inhalation, the researchers found increased MMP-9 expression and increased values of SDC-1 in alveolar lavage fluid from mice, where impaired intercellular junctions were observed. In addition, they also demonstrated through in vivo experiments that downregulation of MMP-9 expression helped protect the glycocalyx, as evidenced by an increase in the glycocalyx of the alveolar epithelium and a decrease in markers of injury in the lavage fluid [53]. In inflammatory environments, MMP-9 is a highly expressed member of the MMP family [54]. Its role as a signaling molecule also regulates the secretion of immunoreactive substances by leukocytes. Although the mechanism by which MMP-9 damages the glycocalyx is not clear, some possible pathways are under investigation. In LPS-related ALI, the researchers observed a reduction in the count of M1-type macrophages along with a decrease in MMP-9 expression as well as a decrease in mortality in the CLP model after using an HIV protease inhibitor. Then, they knocked down MMP-9 expression and found a reduction in M1-type macrophages but an elevation of M2-type macrophages. Finally, they added recombinant MMP-9 and found that the above effects were reversed [55]. This suggests that MMP-9, although secreted by macrophages in an inflammatory setting, can also perform a key function in macrophage phenotypic switching. Therefore, MMP-9 may be associated with inflammatory damage to the endothelium by regulating the type of macrophage polarization in sepsis.

In sepsis, macrophages exert immunobactericidal effects dependent on reactive oxygen species (ROS); furthermore, macrophages are the main source of ROS in ARDS [56]. ROS is produced by NADPH oxidase (Nox). Of these oxidases, Nox2 is the predominant source [57]. ROS is a collective term for a class of oxidizing molecules and groups in the body, commonly including superoxide anion radicals, hypohalous acids, hydroxyl radicals, and hydrogen peroxide. In a homeostatic environment, ROS regulates vascular tone and participates in oxygenation [58]. However, excessive secretion of ROS simultaneously damages endothelial cells. In the state of sepsis, ischemia and reperfusion often occur in various organs of the body. In the course of hepatic ischemia–reperfusion, the first step is the massive death of hepatocytes stimulated by the accumulation of ROS. Concomitantly, cell disintegration products, including damage-associated molecular patterns (DAMPs), enter the circulation, causing macrophages in the liver to release chemokines to recruit more leukocytes and release more ROS [59]. Some studies have shown that ROS-mediated glycocalyx damage may be associated with the activation of Nox2 and xanthine oxidase on the endothelial cell surface [60,61]. Some reports found that Nox2 expression on the endothelial cell surface is upregulated due to inflammatory mediators, and their ROS content is increased compared to controls. Furthermore, by knocking down Nox2, this effect was suppressed [62,63]. With further studies, apart from the directly associated effects of ROS, several indirect mechanisms are gradually being explored. It has been indicated that small amounts of ROS may act as some kind of signaling molecule to degrade the glycocalyx by activating heparinase [64].

In addition to the above permeability factors, shedding of the glycocalyx further exposes the numerous adhesion molecules at the endothelial surface, leading to neutrophil aggregation and adhesion, which eventually will also result in enhanced permeability.

#### 3.2.2. Impaired Endothelial Cell Junctions

Apart from the glycocalyx, several intercellular connections between endothelial cells perform an essential function in the modulation of permeability. The connections between endothelial cells consist of adherens junctions, tight junctions, and gap junctions [65]. In the physiological state, small molecule solutes can selectively reach tissues through these paracellular pathways composed of intercellular junctions. In contrast, large molecules such as albumin and immunoglobulins reach the tissues in the form of vesicles through the caveolae [66]. Albumin recognizes albumin-binding proteins on the surface of endothelial cells, which then causes receptor clustering and caveolae formation (like cell membrane-infolding depression). The caveolae carry albumin across the cell to the outside of the lumen. The other two types of junctions have a similar zipper-like structure and are engaged in the formation of endothelial cell barrier structures [65,67]. However, these two connections differ in the distribution of blood vessels. Tight junctions are predominantly located in small arteries, whereas adherent junctions are found mainly in postcapillary microvessels, which are the main vascular leakage sites in inflammatory environments such as sepsis [67]. Adhesion junctions consist mainly of VE-cadherin, p120, and α- and β-catenin and are connected to actin in the cytoskeleton. Many studies have revealed that VE-cadherin has a significant function during the formation of the vascular endothelium [65]. Reduced intercellular adherens junctions, increased endothelial permeability, and increased neutrophil migration were observed after a VE-cadherin neutralizing antibody was administered to cultured endothelium [68]. Vascular endothelial growth factors (VEGFs) belong to a class of signaling molecules that promote angiogenesis and enhance the permeability of vascular endothelial cells and can be generated by a diverse range of cells, including fibroblasts, the vascular endothelium, and macrophages [69]. Macrophages are able to produce massive levels of VEGFs due to LPS, and elevated circulating levels of VEGFs have been assayed in septic ARDS patients [70]. This has also been evidenced in septic mouse models [71]. These suggest that VEGFs might play an essential part during the process of sepsis development. It has been proposed that VEGFs may enhance β-arrestin 2-regulated endocytosis and thereby internalize more VE-cadherin. In this signaling pathway, the VEGF induces activation of Rac1 and p21 kinases through c-Src-dependent phosphorylation of Vav2 (a guanine nucleotide exchange factor). The activated p21 kinase ultimately causes phosphorylation of VE-cadherin site-specific amino acids and triggers endocytosis [72]. In addition, VEGF downregulates the combination of p120-conjugated proteins with VE-cadherin, which further promotes the endocytosis of VE-cadherin by clathrin [73]. It has also been shown that the pathway involved in the FA kinase intersects with that of the VEGF [74]. The FA kinase enhances the phosphorylation of VE-cadherin and upregulates the expression of c-Src at adherens junctions [75]. Meanwhile, it also promotes the phosphorylation of β-arrestin 2 at site Y142, accelerating its separation from VE-cadherin, which in turn increases endothelial cell permeability [74].

#### 3.2.3. Impaired Endothelial Cytoskeleton

The remodeling of the vascular endothelial cytoskeleton under the influence of pathogens, endotoxins, pro-inflammatory factors secreted by various immune cells, and mechanical stimuli is considered to be deeply engaged in changes in endothelial cell permeability. A normal cytoskeleton maintains the volume of endothelial cells and keeps them tightly arranged. The remodeling of the cytoskeleton is capable of enlarging the paracellular gap, resulting in enhanced permeability [76]. Several subgroups of the Rho family of small GTPases have diverse impacts on cytoskeletal reorganization and endothelial cell permeability [77]. Among them, RhoA is mainly involved in enhancing endothelial cell contraction and increasing permeability. Rac1 and Cdc42, in contrast, facilitate the recovery of the vascular endothelial barrier structure [78]. The underlying mechanism may be that RhoA increases myosin light chain (MLC) phosphorylation and promotes actin-myosin contractility and actin stress fiber formation [79]. In LPS-induced ALI, by administering the anti-inflammatory Schisandrin A, the researchers found that disruption of VE-cadherin and tight junction proteins was reversed. However, the upregulation of RhoA expression diminished the protective benefit of Schisandrin A. They suggest that Schisandrin A may downregulate the RhoA/ROCK1 (RhoA-associated kinase)/MLC signaling pathway and thus reduce endothelial permeability [80]. Because actin and VE-cadherin in the cytoskeleton indirectly form a complex, RhoA may also contribute to the destabilization of adhesion junctions by this mechanism. In addition to this, several studies have shown that activated RhoA is also associated with other signaling transductions that disrupt the endothelial barrier. The function of TNF-α secreted by macrophages to promote endothelial apoptosis and stress fiber formation is regulated by RhoA [81]. Furthermore, RhoA is associated with pathways of endothelial destabilization and increased permeability by LPS, VEGF, histamine, and transforming growth factor-β (TGF-β) [76]. Apart from the Rho family, studies on ROS have shown that they can also cause cytoskeletal remodeling. Under the influence of hydrogen peroxide, the morphology of endothelial cells changed and cell gaps became larger [82]. The induced effects of ROS oxidative stress may also be associated with actin stress fiber formation and cortical actin damage [83]. The endothelium can be stimulated by hydrogen peroxide, and thin filament proteins can be rapidly translocated in the membrane skeleton and cytoplasm, causing the formation of endothelial gaps [84]. Moreover, in a prolonged oxidative stress environment, the reduction in ATP also induced damage and depolymerization of actin microfilaments and microtubules, increasing endothelial permeability [85].

The phenomenon that ROS and RhoA alter endothelial permeability via effects on the cytoskeleton indicates that these two pathways may interact. It has been found that RhoA, Rac1, and Cdc 42 of the Rho family share common redox-sensitive sequences and that ROS are able to bind to them, resulting in the activation of Rho GTPases [86]. Notably, RhoA has one more redox binding site than the other two enzymes, suggesting that it may be more functional. In a study exploring the interaction of ROS and RhoA, LPS stimulation caused an increase in ROS in endothelial cells leading to microtubule disassembly. In turn, the RhoA activator GEF-H1 was released from microtubules, further activating RhoA [87]. Another study found that histone deacetylase 6 (HDAC 6) mediated microtubule disintegration [88]. This suggests a possible upstream–downstream regulatory relationship between ROS, HDAC6, microtubules, and RhoA.

#### 3.2.4. Enhanced Endothelial Cell Apoptosis

Apoptosis is one of the ways in which homeostasis is regulated in the organism and is also known as programmed death. In the physiological state, endothelial cells accomplish self-regulation through apoptosis in small percentages. In the setting of sepsis, many endothelial cells are heading toward apoptosis [89]. The septic cascade response produces high levels of factors that might cause apoptosis in endothelial cells, including TNF-α, interferon, and ROS [90,91]. Essentially all of these signaling molecules can be produced by activated macrophages, further suggesting the complexity of the influence between macrophages and endothelial cells. The interaction between endothelial cells and various types of cells on the lumen side may further upregulate pro-apoptotic signaling. For example, in the setting of LPS, induced monocytes enhance programmed apoptosis in endothelial cells through TNF-α-mediated mechanisms [92]. It has been found that endothelial apoptosis directly stimulated by TNF-α is closely related to caspases and p38 mitogen-activated protein kinase (MAPK) [93]. Nitric oxide (NO) is a signaling molecule that originates from endothelial cells under physiological conditions via endothelium-derived nitric oxide synthase (eNOS), which has the ability to reduce platelet and leukocyte adhesion and dilate blood vessels [94]. At low levels, NO inhibits endothelial cells moving toward apoptosis (e.g., TNF-α stimulation), whereas high concentrations can lead to apoptosis [95]. Activated macrophages are able to generate large quantities of NO via inducible nitric oxide synthase (iNOS) [96]. NO can generate peroxynitrite by reacting with other superoxide anions. As a strong oxidant, it can oxidize DNA bases and modify lipids and proteins. More importantly, it leads to impaired mitochondrial function by directly damaging the mitochondrial membrane and disrupting the electron transport chain [97]. Ultimately, the apoptotic response is initiated by the combined effect of reduced ATP and increased ROS [98]. Selective iNOS inhibition alleviates sepsis-related renal insufficiency and increases survival in some animal studies [99,100].

### 3.3. Macrophages Enhance Endothelial Cell Procoagulant Capacity

In healthy conditions, the intact vascular endothelium is free from thrombus and platelet aggregation. Maintenance of this anticoagulant effect depends on endothelium-derived NO, which prevents platelet activation and adhesion [94]. In addition, the smooth endothelial barrier on the surface of blood vessels reduces the contact between endothelial cells and the various clotting components of the blood [14]. However, as mentioned previously, in the microenvironment of sepsis, disruption of the glycocalyx increases endothelial cell interactions with a variety of coagulation factors and predisposes one to coagulation disorders.

Disseminated intravascular (DIC) coagulation is the most severe form of coagulation disorder. DIC is one of the serious complications of sepsis, with a high mortality rate and poor patient prognosis. The incidence of DIC in severely septic patients can be 30% to 50% [101]. Characteristics of DIC include impairment of endothelial function and imbalances in the coagulation, anticoagulation, and fibrinolytic systems, which are manifested by the formation of widespread microthrombi in the microvasculature [102]. Macrophages also perform an active function in this pathologic process.

TNF-α is a major inflammatory molecule in sepsis [103] and regulates endothelial coagulation in addition to affecting endothelial permeability. Many studies have demonstrated that TNF-α is able to downregulate eNOS expression [104,105]. The reduction in eNOS subsequently impairs the anticoagulant effect of endothelial-derived NO. Researchers injected mice with exogenous TNF-α and found that the mice developed a sepsis-like hypercoagulable state. At the same time, the researchers treated mice with amitriptyline and found that the CLP mice had reduced levels of TNF-α and suppressed M1-type polarization of macrophages, along with attenuated coagulation [106]. Amitriptyline is an inhibitor of acid sphingomyelinase. In addition, it was also found that glomerular endothelial cell damage was alleviated in mice with acute kidney injury intervened with amitriptyline [107]. This may be due to its inhibition of macrophage-derived inflammatory exosome production. Similarly, it has been discovered that blocking macrophage-secreted interleukin-1 also improves survival in DIC in sepsis [108].

Tissue factor (TF) is the starting point of the exogenous coagulation pathway. TF is released from the vascular endothelium injured by pathogen-associated molecular patterns (PAMPs) and inflammatory molecules and then binds to coagulation factor 12 to initiate the exogenous coagulation pathway [101]. However, recent studies have indicated that TF produced by monocytes and macrophages is responsible for pathological coagulation in sepsis instead of endothelial sources [109,110]. TF expression may be related to the LPS-TLR4-NF-κB pathway in monocytes [110]. In addition, it has been discovered that monocytes probably release TF in the form of particles [111]. Another study found that the upregulated expression of TF in monocytes and macrophages was associated with the ROS-SENP3-JNK pathway [112].
biomolecules-14-00980-t001_Table 1Table 1Overview of molecules secreted by macrophages. We list several inflammatory molecules secreted by macrophages that are mainly discussed. The table shows the individual molecules affecting some functions of endothelial cells and possible mechanisms.Molecules Secreted by MacrophagesAffectedFunctionsPossible MechanismsReferencesCCL2adhesionPositive feedback effect withendothelial cells[21,23,26]TNF-αadhesionActivates ZIPK/ NF-κB[31]permeabilityActivates heparanase;upregulates MMP-9[50,51]coagulabilityDownregulates eNOS expression[103,104]ROSpermeabilityUpregulates Nox2;activates RhoA/ROCK1/MLC[59,60,82,85]VEGFpermeabilityPromotes FA kinasephosphorylation of VE-cadherin[74]MMP-9permeabilityRegulates macrophagepolarization[54]NOpermeabilityImpairs mitochondrial energyproduction[96]TFcoagulabilityInitiate exogenous coagulation pathway[100]


## 4. Potential Therapeutic Target—Modulation of Macrophage Polarization

Various studies from the previous section have shown that many kinds of molecules secreted by macrophages enhance endothelial cell adhesion and permeability through various pathways. Although modulation of the upstream and downstream pathways of these factors can have a protective effect on the endothelium, more and more research is focusing on macrophages, the source of inflammatory factors. In general, macrophages, which are at the forefront of immunity, are activated by LPS. LPS induces macrophage polarization by binding to TLR4 and then activating myeloid differentiation protein 88 (MyD 88), which ultimately activates NF-κB, triggering the secretion of a large number of cytokines including TNF-α and iNOS [113]. These substances enhance the adhesion and permeability of endothelial cells. This activation pattern is known as PAMP-dependent because it requires the involvement of pathogen components. At the same time, some substances secreted by lymphocytes are involved in macrophage activation, such as interferon (IFN)-γ. It is able to activate resting macrophages into classically activated M1 pro-inflammatory cells. Th1 (helper T-cell) cells are able to secrete substances (IFN-γ and TNF-α) that promote macrophage differentiation to the M1 type, whereas Th2 cells (secreting IL-4 and IL-13) promote the differentiation of M2-type macrophages [114]. M2-type macrophages can generate anti-inflammatory TGF-β and IL-10 [115]. TGF-β promotes the recruitment of hematopoietic cells and increases their VEGF expression, triggering angiogenesis and vascular remodeling to promote tissue repair in inflammation [115]. The homeostatic maintenance of the immune system requires that the two polarized cells be in equilibrium. In the microenvironment of sepsis, dysregulation of the immune response caused by an imbalance between the two polarizations is a key mechanism of sepsis. M1-type macrophages predominate, triggering excessive inflammatory responses and alterations in endothelial cell adhesion and permeability in the early phase of sepsis. In contrast, M2-type macrophages have the opposite effect. Therefore, regulation of macrophage polarization is a potential target for the treatment of endothelial cell dysfunction, as shown in Table 2.

### 4.1. Regulation of Macrophage Polarization by miRNA

MicroRNAs are defined as short-stranded (~20 bases) non-coding nucleic acid sequences that are mainly responsible for silencing mRNAs and repressing targeted gene expression [116]. Their role in epigenetic inheritance is receiving increased attention and has been shown to have regulatory roles in inflammation and macrophage polarization [117]. The TLR4-NF-κB pathway contributes to polarization in M1-type macrophages [118]. A significant reduction in miR-164a was assayed in septic patients [119]. After overexpression of miR-164a, the TLR4-NF-κB pathway was inhibited, and sepsis-related myocardial damage was attenuated, as evidenced by reduced production of inflammatory cells and inflammatory factors [119]. This suggests that miR-164a inhibits M1-type macrophage production and thereby reduces unfavorable changes in endothelial cell permeability and adhesion.

A recent study summarizes the multiple ways in which exosome-wrapped miRNAs affect macrophage polarization in sepsis [120]. For example, after pretreatment of MSCs (mesenchymal stem cells) with IL-1 β, the investigators obtained exosomes containing miR-21 secreted by MSCs. Injecting the exosomes into CLP septic mice, an increase in the number of macrophages polarized towards the M2 type was observed, and the mice showed improved symptoms and survival [121]. This study implies that the immunomodulatory properties of MSC may be related to miRNA, and the combination of exosomes and miRNA may enhance the regulatory role of miRNA. Thus, induced polarization of macrophages toward the M2-type direction is also a potential way to promote the repair of endothelial cell dysfunction. Another study suggests that the role of miR-21 in promoting M2-type polarization may be related to the colony-stimulating factor-1 receptor (CSF1-R) [122]. Notably, in another study on miR-21, it could in turn promote M1-type polarization by targeting STAT3 [123]. Such seemingly contradictory results illustrate the complexity of miRNA regulation. However, it has also been found that some miRNA can exacerbate ALI by promoting M1 macrophage polarization. MiR-23a-3p is able to activate M1 macrophages and target polo-like kinase 1 (PLK1) through the STAT1/STAT3 pathway [124]. Therefore, inhibition of the expression of such miRNAs could also be considered to reduce the proportion of M1-type macrophages. Currently, miRNA species and subclassifications are very diverse and need to be continuously screened to form their regulatory networks affecting macrophage polarization. Existing studies are mostly confined to animal and cellular experiments and need to be expanded to reach clinical trials.

### 4.2. Regulation of Macrophage Polarization by Medicines

Several reports suggest some natural components of Chinese herbs have anti-inflammatory and macrophage-polarizing effects. Luteolin is a class of flavonoid compounds derived from plants. After co-culturing it with IL-4-induced M2 macrophages and LPS-induced M1 macrophages, the secretion of cytokines (IL-6 and TNF-α) and surface markers was found to be reduced in M1, whereas a substantial elevation in the relevant indicator (IL-10) was detected in M2. Further, it has been found that the mechanism for this altered polarization may be the inhibition of p-STAT3 and enhancement of p-STAT6 [125]. An active ingredient called 1,3,6,7-tetrahydroxy-8-prenylxanthone (TPX) can be isolated from mangosteen. It has been found that TPX inhibits nuclear translocation of p65 and activation of NF-κB in macrophages in the setting of LPS, which ultimately reduces downstream expression of iNOS, COX2, and TNF-α. Finally, a decrease in M1-type macrophage markers but an increase in M2-type markers in response to TPX intervention has also been found [126]. The decline in pro-inflammatory cells and factors observed in the above herbs or active ingredients in them could provide an important basis for reversing endothelial cell adhesion and permeability in sepsis.

In addition to these traditional Chinese medicines, drugs used clinically for other diseases have also been found to be involved in macrophage polarization processes. Canagliflozin (CANA) is a sodium-glucose cotransporter protein 2 inhibitor commonly found in type 2 diabetes, and it also has anti-inflammatory properties [127]. In a recent study, it was found to elevate the M2/M1 proportion and reduce the secretion of inflammatory factors in mice with LPS-induced ALI [128]. In addition to this, liraglutide can also modulate macrophage polarization [129]. Considering their original pharmacological mechanisms, we speculate that they may be associated with reprogramming of macrophage glucose metabolism, which in turn impacts polarization patterns. Furthermore, researchers have found that patients with high glucose levels have a higher chance of developing severe COVID-19 pneumonia. The high-glucose environment promotes the release of inflammatory molecules from macrophages via the ROS-HIF-1a (hypoxia-inducible factor-1a)/Glycolysis axis while suppressing adaptive immunity. This may explain why patients with high glucose levels are prone to develop more severe viral lung injury [130]. It has also been noted that M1-type polarization of macrophages was inhibited by decreasing the glycolytic response [131]. In addition to glycolysis, nutrient metabolism, such as fat and amino acids, is also closely involved [114]. This further illustrates the impact of energy metabolism reprogramming on the functional state of macrophages. The regulation of this process is of great research value. Given that natural herbal compounds and other clinical agents have multiple modulatory effects on polarization, the pathways and modalities associated therein remain incompletely understood. Therefore, macrophage polarization and related treatment compounds ought to be studied in greater detail to determine the best strategies to target in order to ameliorate endothelial cell dysfunction in sepsis.
biomolecules-14-00980-t002_Table 2Table 2Modulation of macrophage polarization. We list some of the miRNAs regulating the direction of macrophage polarization and the potential pathways. In addition, some Chinese herbs and clinically used drugs have been found to have similar effects and to alleviate sepsis damage. By interfering with macrophage polarization, reducing the production of inflammatory factor-secreting M1-type macrophages and increasing the proportion of M2-type macrophages, endothelial cell function may be restored in this way.Substance of InterventionM1-Type MacrophageM2-Type MacrophagePotential Signaling Pathwayor Metabolic PathwayReferenceYearmiR-164a↓
TLR4-NF-κB[119]2015miR-21
↑CSF1-R[122]2015↑
STAT3[123]2015miR-23a-3p↑
PLK1-STAT1/STAT3[124]2022Luteolin↓
p-STAT3[125]2020
↑p-STAT6TPX↓↑p65 and NF-κB[126]2018Canagliflozin↓↑glucose metabolism[127]2019Liraglutide[129]2022↑: Activate/upregulate ↓: Inhibit/downregulate.


## 5. Conclusions

Endothelial dysfunction lies at the critical point of organ damage in sepsis and can lead to hypoperfusion and decreased oxygenation. Various molecules secreted by macrophages are the source of endothelial cell damage. An in-depth study of how these factors affect endothelial cell function and structure will help develop therapeutic strategies to protect vascular barrier stability. In the microenvironment of sepsis, macrophages subjected to various stimuli further differentiate into M1- and M2-polarized types of cells that exert pro-inflammatory damage and anti-inflammatory repair. Therefore, promoting their polarization in a certain direction or maintaining the balance of macrophage polarization at an appropriate stage is also a promising therapeutic target for endothelial cell protection.

## Figures and Tables

**Figure 1 biomolecules-14-00980-f001:**
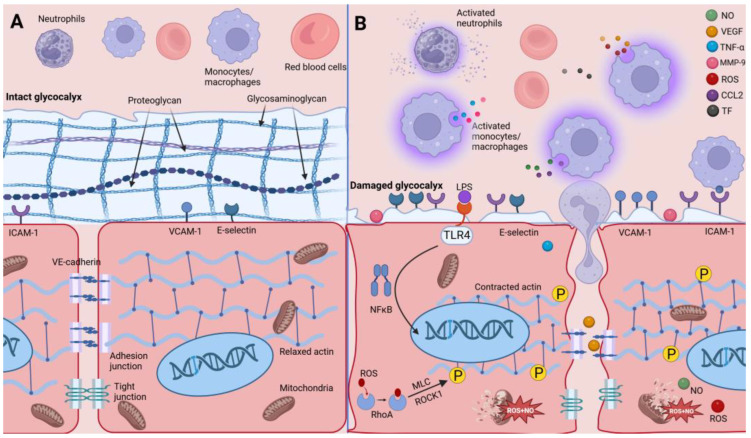
Comparison of endothelial cells under physiologic and pathologic conditions. (**A**) In the physiologic state, blood cells are in a relatively quiescent state. Endothelial cells are in a tightly packed state containing an intact surface glycocalyx, robust intercellular junctions, a relaxed cytoskeleton, and small amounts of adhesion molecules and selectins. (**B**) In the setting of LPS or sepsis, monocytes, macrophages, and neutrophils produce a large number of signaling molecules, including NO, VEGF, TNF-α, MMP-9, ROS, TF, and CCL-2. These molecules lead to structural and functional alterations in endothelial cells, including degradation of the glycocalyx, increased exposure to and expression of adhesion molecules and selectins, rupture of intercellular connections, and cytoskeleton shrinkage by phosphorylation, and apoptosis due to mitochondrial disruption. Neutrophils cross the endothelial cell barrier through an enlarged paracellular gap. These changes ultimately cause an increase in the adhesion and permeability of the endothelium.

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
