# Peer review of "Endothelial Cell Dysfunction Due to Molecules Secreted by Macrophages in Sepsis"

_biomolecules, 2024, doi:10.3390/biom14080980_

Round 1

Reviewer 1 Report

Comments and Suggestions for Authors

See Review

Reviewer 2 Report

Comments and Suggestions for Authors

This review summarizes well the changes in endothelial cell function in sepsis and the contribution of macrophages in it.  Several comments are noted, as well as the following typos.

The author sometimes refers to "leukocytes" and the specific cell type is unclear (e.g., line 115). The cell type should be specified.

Line 118 Receptors for NF-kB is misleading

Line 130 Specify the type of mononuclear cells. Does it include lymphocytes? 

Line 156 passage from within ?

Line 215 edema, The -> edema, the

Line 255 DAMP should be with its full name

Line 263 It has been suggested that indicated that

Line 272 Endotheilal -> endothelial

Line 278 Describe more details about transport of large molecules via the caveolae pathway

Line 290 Endothelium -> endothelium

Line 316 several -> Several

Line 408 Th1 and Th2 are not substance. They are types of helper T cell.

Line 441 MSCs should be written with its full name

Comments on the Quality of English Language

English is good.
